# Factors Associated with Ocular and Extraocular Recovery in 143 Patients with Sarcoid Uveitis

**DOI:** 10.3390/jcm9123894

**Published:** 2020-11-30

**Authors:** Francois-Henri Bienvenu, Théophile Tiffet, Delphine Maucort-Boulch, Mathieu Gerfaud-Valentin, Laurent Kodjikian, Laurent Perard, Carole Burillon, Cécile-Audrey Durel, Arnaud Hot, Yvan Jamilloux, Pascal Seve

**Affiliations:** 1Department of Internal Medicine, Hôpital de la Croix-Rousse, Université Claude Bernard Lyon 1, 69004 Lyon, France; francois-henri.bienvenu@chu-lyon.fr (F.-H.B.); mathieu.gerfaud-valentin@chu-lyon.fr (M.G.-V.); yvan.jamilloux@chu-lyon.fr (Y.J.); 2Hospices Civils de Lyon, Pôle Santé Publique, Service de Biostatistique et Bioinformatique, F-69003 Lyon, France; theophile.tiffet@protonmail.com (T.T.); delphine.maucort-boulch@chu-lyon.fr (D.M.-B.); 3CNRS, UMR 5558, Laboratoire de Biométrie et Biologie Évolutive, Équipe Biostatistique-Santé, F-69100 Villeurbanne, France; 4Department of Ophthalmology, Hôpital de la Croix-Rousse, Université Claude Bernard Lyon 1, 69004 Lyon, France; laurent.kodjikian@chu-lyon.fr; 5Laboratoire UMR-CNRS 5510 Matéis, Université Lyon 1, 69100 Villeurbanne, France; 6Department of Internal Medicine, Centre Hospitalier Saint-Joseph Saint-Luc, Université Claude Bernard Lyon 1, 69007 Lyon, France; lperard@ch-stjoseph-stluc-lyon.fr; 7Department of Ophthalmology, Hôpital Edouard Herriot, Université Claude Bernard Lyon 1, 69003 Lyon, France; carole.burillon@chu-lyon.fr; 8Department of Internal Medicine, Hôpital Edouard Herriot, Université Claude Bernard Lyon 1, 69003 Lyon, France; cecile-audrey.durel@chu-lyon.fr (C.-A.D.); arnaud.hot@chu-lyon.fr (A.H.); 9Hospices Civils de Lyon, Pôle IMER, Lyon F-69003, France

**Keywords:** uveitis, sarcoidosis recovery

## Abstract

Background: Sarcoidosis is one of the leading causes of uveitis. To date, no studies have assessed the factors specifically related with recovery in ocular sarcoidosis. In this study, we aimed to determine factors associated with ocular and extraocular recovery in patients with sarcoid uveitis. Methods: A retrospective study of sarcoid uveitis, with a three-year minimum follow-up in Lyon University Hospital between December 2003 and December 2019. Patients presented biopsy-proven sarcoidosis or presumed sarcoid. Recovery was defined by a disease-free status, spontaneously or despite being off all treatments for three years or more. Results: 143 patients were included: 110 with biopsy-proven and 33 with presumed sarcoid uveitis. Seventy-one percent were women, the median age at presentation was 53 years, and 71% were Caucasian. Chronic uveitis was the main clinical presentation (75%), mostly panuveitis (48%) with bilateral involvement (82%). After a median follow-up of 83.5 months, recovery was reported in 26% of patients. In multivariable analysis, Caucasian ethnicity (*p* = 0.007) and anterior uveitis (*p* = 0.008) were significantly associated with recovery, while increased intraocular pressure was negatively associated (*p* = 0.039). Conclusion: In this large European cohort, one quarter of patients recovered. Caucasian ethnicity and anterior uveitis are associated with ocular and extraocular recovery.

## 1. Introduction

Sarcoidosis is a systemic inflammatory disease of unknown etiology, characterized by noncaseating epithelioid cell granuloma infiltrates in miscellaneous organs [1]. Lungs and lymphatics are frequently affected; however, any organ can be involved. Ocular manifestations are one of the most common extrathoracic manifestations of sarcoidosis [2]. Although all ocular structures can be involved, uveitis is the most common presentation occurring in 25–50% of sarcoidosis patients [2]. Uveitis can be the revealing manifestation of sarcoidosis and is mainly diagnosed within the first year after disease onset [2,3]. Two peaks of incidence are reported: the first one in patients aged 20 to 30 years, and the second in those aged 50 to 60 years (mostly in Caucasian female patients) [1,2].

Sarcoid uveitis is responsible for severe visual impairment in 2.4–10% defined as best-corrected visual acuity, (BCVA) < 20/200 in at least one eye [4,5,6]. Visual loss is mostly related to cystoid macular edema [4,7,8]. Factors associated with poor visual outcome are late-age onset, black phototype, female gender, underlying chronic systemic disease, ocular posterior segment involvement, chronic cystoid macular edema, multifocal choroiditis, persistent ocular inflammation, and glaucoma [2,4,8,9,10,11].

In sarcoidosis, the factors that have been associated with recovery are an age <40 years at onset, Caucasian ethnicity, acute uveitis, and a pulmonary radiologic stage I [12]. To date, no study has assessed the factors specifically associated with recovery in sarcoid uveitis. Thus, the objective of this study was to determine the factors associated with ocular and extraocular recovery in patients presenting with sarcoid uveitis.

## 2. Experimental Section

### 2.1. Study Population

Medical records of patients aged ≥18 years of two departments of Ophthalmology and Internal Medicine at Lyon University Hospitals diagnosed with biopsy-proven or presumed sarcoid uveitis between December 2003 and December 2019 were retrospectively reviewed. Only patients who had a minimal follow-up duration of 3 years were included.

Biopsy-proven sarcoidosis was defined according to the World Association for Sarcoidosis and Other Granulomatous Disorders/American Thoracic Society/European Respiratory Society (WASOG/ATS/ERS) criteria [1]. Sarcoidosis was presumed when two or more of the following criteria were met: abnormal chest imaging compatible with sarcoidosis, elevated serum angiotensin-converting enzyme (ACE), predominantly CD4 lymphocytic alveolitis (presence of >15% lymphocytes and CD4/CD8 ratio >3.5 in bronchoalveolar lavage (BAL) fluid analysis) [1].

Patients diagnosed with other granulomatosis (e.g., tuberculosis) and patients who had a positive treponemal antibody test were not included.

Complementary investigations were realized according to clinical examination and ocular findings in order to exclude differential diagnosis [13]. Ethnicity was defined according to the patient’s origin: Caucasian (white-European), North-African (Moroccans, Algerians, Tunisians) and Afro-Caribbean (sub-Saharan region and French West Indies).

This study received approval from the local ethics committee in February 2019 (No. 19–31) and was registered on clinicaltrials.gov (NCT 03863782).

### 2.2. Clinical Data

A standardized form was used to collect and analyze demographic, clinical and ophthalmologic data. Ophthalmologists were responsible for ophthalmologic data collection (i.e., findings of examination of the eyelids and conjunctiva, cornea, anterior chamber, iris, lens, vitreous, and retina) at initial presentation and during follow-up. Standardization of Uveitis Nomenclature (SUN) was used for anatomical classification of uveitis [14].

Internal medicine physicists were responsible for clinical data, dosages of serum ACE and lysozyme, and imaging results gathering. Radiologists reviewed all imaging findings. A positive chest X-ray was defined by bilateral hilar lymphadenopathy (BHL), pulmonary granuloma, or ground-glass parenchymal opacity. Positivity criteria for chest CT were hilar or mediastinal lymph nodes with a short axis diameter of >10 mm, perilymphatic pulmonary nodules, or other parenchymal lung abnormalities characteristic of sarcoidosis [15]. A ^18^F-labelled fluorodeoxyglucose positron emission tomography (^18^F-FDG PET) was considered positive when it showed increased FDG uptake (higher than the uptake of the adjacent normal tissue) in the mediastinum and/or lung parenchyma or extrathoracic localizations [16]. Systemic sarcoidosis was defined by clinical or biopsy evidence of extraocular sarcoidosis within 1 year after the onset of uveitis in contrast to isolated ocular form as previously described [4]. Extraocular extent of the disease throughout follow-up was defined as new symptomatic sarcoidosis in a noninvolved organ at baseline and that occurred after a minimum of 1 year after uveitis diagnosis [17]. Local corticosteroids (drops, periocular injection and intravitreal injection) and systemic treatments (oral corticosteroids and immunosuppressants) were given according to the laterality (unilateral or bilateral), the localization of uveitis and extraocular involvement. Biologic treatments were used according to French guidelines [18].

### 2.3. Outcomes

The main outcome was recovery defined as a disease-free status (ocular and extraocular), without any kind of treatment for 3 years or more [19].

### 2.4. Statistical Analysis

Data are described as frequencies and percentages for categorical variables and as medians (25th–75th percentile range) for quantitative variables. In order to describe the association between recovery and its predictors, we used a multivariate logistic regression. Variables at diagnosis used for analysis were sex, age >50 years, ethnicity (Caucasian vs. others), gender, chronicity, anatomical localization (anterior vs. others), granulomatous uveitis, synechia, increased IOP (defined as chronically elevated intraocular pressure >21 mmHg resulting from damage of angle), vasculitis, chronic macular edema, papillary edema, peripherical multifocal choroiditis, extraocular involvement, abnormal chest imaging (X-ray and/or computerized tomography (CT)) and increased serum ACE. A stepwise elimination using the Akaike’s information criterion (AIC), was applied to derive the most fitting and most parsimonious model [20]. Models before and after stepwise elimination were compared using an analysis of variance. For all statistical analyses, *p* < 0.05 was considered significant. Analyses were performed with R-Software, V.3.1.1 (R Foundation for Statistical Computing, Vienna, Austria).

## 3. Results

### 3.1. General Characteristics at Baseline

A total of 143 patients were included; 71% were women (*n* = 101). Median age at presentation was 53 (38–64) years. Median follow-up was 84 (58–124) months, 71% of the patients were Caucasians and 77% of the patients had a biopsy-proven sarcoidosis.

Uveitis revealed sarcoidosis in 114 patients (80%), and 53 (37%) patients had extraocular sarcoidosis involvement at the time of diagnosis (Table 1).

### 3.2. Ocular Findings at Baseline

Uveitis was chronic in 108 patients (76%) and bilateral in 117 patients (82%). Panuveitis was the most frequently observed type of uveitis, occurring in 48% of the patients, 31 patients (22%) with increased IOP. Of note, 54 patients had peripherical multifocal choroiditis (38%). Vasculitis was diagnosed in 29% of the patients and was occlusive in 3/41(7%). Chronic macular edema was noted in 69 patients (48%) (Table 2).

### 3.3. Extent of Sarcoidosis during Follow-Up

For 27 patients (19%), sarcoidosis involvement extended during follow-up, among them 15 patients had isolated ocular sarcoidosis at diagnosis. Median delay for extent was 36 (20; 69) months. Six patients presented more than one additional involvement. Additional involvements were mostly pulmonary (*n* = 9) and cutaneous (*n* = 9) (Table 3).

#### 3.3.1. Treatments and Outcomes

Note that 133 patients (93%) were treated with local corticosteroids. Systemic treatment was required in 114 patients (80%) (Table 3). Oral corticosteroids were used in 109 patients (76%) and immunosuppressive drugs in 75 patients (52%). The median duration of oral corticosteroids and immunosuppressive drugs was, respectively, 42 (24–79) and 34 (17–52) months. Ocular condition was the main indication for oral corticosteroids (64%) and immunosuppressants (52%). Oral corticosteroids dependency (defined as >10 mg of prednisone/day) was reported in 20% of cases. Visual outcome was favorable with 94% of patients showing a BCVA > 20/50. Vitrectomy was necessary for epiretinal membrane (ERM) secondary to ocular inflammation in 14 patients (9.8%) (Table 4).

#### 3.3.2. Main Outcome

Recovery was reported in 37 patients (26%). Among these patients, 65% presented isolated sarcoid uveitis at onset and during follow-up (Table 4). In multivariate analysis, Caucasian ethnicity (OR: 5.32, CI 95% (1.73–20.35), *p* = 0.007) and anterior uveitis (OR: 5.15, CI 95% (1.6–18.85), *p* = 0.008) were associated with recovery, whereas increased IOP was negatively associated with recovery (OR: 0.28, CI 95% (0.07–0.86), *p* = 0.039) (Table 5).

## 4. Discussion

We reported a large cohort of patients with an extended follow-up (mean duration >7 years) focusing on ocular and extraocular recovery in sarcoid uveitis. Most of the patients had biopsy-proven sarcoidosis, and exclusion of other granulomatous diseases was required for inclusion to increase its relevance. International Workshop on Ocular Sarcoidosis (IWOS) criteria were voluntary not used due to their lack of sensitivity [21]. These criteria were recently actualized, and their diagnostic performance has only been evaluated in one study to date [22,23].

Our population is predominantly Caucasian middle-aged females, consistent with sarcoid uveitis in general [24]. Ophthalmologic features were similar to those of previous cohorts with a predominance of bilateral, chronic and granulomatous uveitis [21,24,25]. We mainly described panuveitis, although it is reported in 9–30% of cases in literature [26].

Sarcoidosis prognosis is variable according to initial presentation, localization, ethnicity and age at onset [12]. Recovery definition was based on the fact that sarcoidosis rarely relapses after three years without symptoms and treatment withdrawal (mostly corticosteroids) [27]. Our results show that about a quarter of patients recover. Caucasian ethnicity and anterior uveitis are associated with better chances of ocular recovery.

Caucasian ethnicity is known to be overall associated with a better prognosis in sarcoidosis [28,29,30]. Concerning visual outcome, we recently conducted a study showing that white European patients with sarcoid uveitis tend to have more chronic macular edema and chronic uveitis, and thus worser visual outcome in comparison with other ethnic groups [17]. However, these patients were less likely to present extraocular involvement. We hypothesize that this characteristic could partially explain a better chance of recovery in these patients due to the frequent necessity of treatment in systemic sarcoidosis. At diagnosis, patients mainly presented isolated uveitis or mild extraocular involvements; this could potentially explain why this parameter did not influence recovery.

Unlike anterior uveitis, intermediate or posterior uveitis are associated with worse prognosis and often require systemic treatments [26,31]. In our study, most patients with anterior uveitis presented acute uveitis. This subcategory of uveitis is associated with a better overall prognosis [8,9,10,11,32]. The association of anterior uveitis, in comparison to other anatomical localization, with recovery is consistent with previous data.

We also found a negative association with recovery among patients who presented increased IOP at diagnosis. About 9% to 25% of patients with sarcoid uveitis presented glaucoma [3,4]. Glaucoma has been associated with poor visual prognosis in several studies, notably as a complication of anterior or posterior sarcoid uveitis [4,8,9,10,11]. Glaucoma can lead to blindness, especially in African-American patients [3].

Abnormal chest imaging (X-Ray and/or CT) tends to be associated with recovery; however, almost all our patients presented this criterion at diagnosis, explaining the lack of significance with a large confidence interval. Only a few patients had a clinical pulmonary involvement. Almost all patients presented a radiographic stage of 0 or 1. Mild radiologic involvement is conventionally related to acute disease and associated with a better prognosis and higher chances of recovery [12].

Only a few studies have focused on the prognosis of chronic uveitis with an extended follow-up. Most of them are cross-sectional, and the follow-up of visual acuity is thus not standardized [33,34,35,36]. Visual impairment in at least one eye occurs in 25–35% of cases. BCVA > 20/50 in various cohorts is found in 65–90% of patients, and 90% achieve a BCVA of 20/200 or more [33,34,36]. However, the heterogeneity of patients in these studies, including all kinds of uveitis and not only sarcoid uveitis, reduces the value of comparison.

This study has some limitations. First, it was carried out in a tertiary reference center. Thus, severe uveitis (chronic uveitis, panuveitis) was probably overrepresented and mild uveitis was therefore underestimated. Second, because of its retrospective nature, we could not obtain full data from all patients, especially full ophthalmologic descriptions and treatments.

Moreover, the three ethnic samples were unequal with a majority of Caucasian patients and results cannot be generalized to other populations given the local ethnic distribution.

In conclusion, this study reports the factors associated with ocular and extraocular recovery in patients with sarcoid uveitis. Anterior uveitis at diagnosis and Caucasian ethnicity were associated with better chances of recovery. Further studies are needed to assess other potential predictive factors of prognosis in sarcoid uveitis.

## Figures and Tables

**Table 1 jcm-09-03894-t001:** General characteristics at diagnosis of the 143 patients with sarcoid uveitis.

Characteristics	*n* (%) or Median (25th–75th Percentile Range)
Age (years)	53 (38–64)
Female gender	101 (70.6)
Follow-up period (months)	84 (58–124)
Ethnicity	
Caucasian	102 (71.3)
North African	27 (18.9)
Afro-Caribbean	14 (9.8)
Biopsy-proven sarcoidosis	110 (76.9)
Extraocular sarcoidosis	53 (37.1)
Uveitis revealing sarcoidosis	114 (79.7)
Serum markers	
High ACE	83/136 (61.0)
High lysozyme level	84/103 (81.6)
Chest imaging	
Abnormal chest X-ray	74/129 (57.4)
Stage 1	71/74 (95.9)
Stage 2	3/74 (4.1)
Abnormal chest CT	125/140 (89.3)
Abnormal ^18^F-FDG PET	71/86 (82.6)

CT: computerized tomography; ACE: angiotensin-converting enzyme; ^18^F-FDG PET: ^18^F-labelled fluorodeoxyglucose positron-emission tomography.

**Table 2 jcm-09-03894-t002:** Ophthalmologic findings of the 143 patients with sarcoid uveitis.

Characteristics	*n* (%)
Anterior uveitis	27 (18.9)
Anterior + intermediate uveitis	27 (18.9)
Intermediate uveitis	8 (5.6)
Posterior uveitis	13 (9.1)
Panuveitis	68 (47.5)
Bilateral involvement	117 (81.8)
Chronic form	108 (75.5)
Granulomatous uveitis	80 (55.9)
Synechia	58 (40.6)
Increased IOP	31 (21.7)
Peripherical multifocal choroiditis	54 (37.8)
Chronic macular edema	69 (48.3)
Papillary edema	26 (18.2)
Vasculitis	41 (28.7)

IOP: chronically elevated intraocular pressure >21 mm Hg resulting from damage of angle.

**Table 3 jcm-09-03894-t003:** Extent of the sarcoidosis during follow-up of the 143 patients with sarcoid uveitis.

Characteristics	*n* (%) or Median (25th–75th) Percentile Range
Systemic extent	27 (18.8)
Isolated ocular sarcoidosis at diagnosis	15 (10.4)
Extent involving >1 organ	6 (4.1)
Median delay (months)	36 (20–69)
Pulmonary involvement	9 (6.3)
Cutaneous involvement	9 (6.3)
Arthritis	5 (3.5)
Liver involvement	3 (2.1)
Cardiac involvement	3 (2.1)
Renal involvement	1 (0.7)
Hypercalcemia	3 (2.1)

**Table 4 jcm-09-03894-t004:** Treatments and outcomes of the 143 patients with sarcoid uveitis.

Characteristics	*n* (%) or Median (25th–75th Percentile Range)
Local treatment (eye drops or injection)	133 (93.0)
Systemic treatment	114 (79.7)
Oral corticosteroids	109 (76.2)
Immunosuppressants	75 (52.4)
Methotrexate	45 (31.5)
Hydroxychloroquine	30 (21.0)
Anti-TNF	4 (2.8)
Indication for oral corticosteroids	
Ocular	69/109 (63.3)
Extraocular	9/109 (8.3)
Both	31/109 (28.4)
Indication for immunosuppressants	
Ocular	39/75 (52.0)
Extraocular	8/75 (10.7)
Both	28/75 (37.3)
Oral corticosteroid dependency (>10 mg of prednisone/day)	29 (20)
Steroid duration (months)	42 (24–79)
Immunosuppressant duration (months)	34 (17–52)
Systemic therapy at last report	
Oral corticosteroids	55 (38.9)
Mean dose (mg of prednisone/day)	7.4
Immunosuppressants	30 (20.9)
Outcomes	
Ocular and extraocular recovery	37 (25.9)
Isolated ocular sarcoidosis	24/37 (64.9)
BCVA ≤ 20/50	8 (5.6)
BCVA ≤ 20/200	1 (0.6)
Complete visual recovery	65 (45.5)
Vitrectomy for ERM	14 (9.8)

TNF: Tumor Necrosis Factor; BCVA: best-corrected visual acuity in both eyes; ERM: epiretinal membrane.

**Table 5 jcm-09-03894-t005:** Factors associated with ocular and extraocular recovery after stepwise elimination.

Characteristic	Odds Ratio	95% Confidence Interval	*p* Value
Caucasian ethnicity	5.32	1.73–20.35	0.007
Anterior uveitis	5.15	1.60–18.85	0.008
Granulomatous uveitis	2.28	0.97–5.62	0.064
Increased IOP	0.28	0.07–0.86	0.039
Abnormal chest imaging (X-Ray or CT)	9.60	1.43–201.24	0.052

CT: Computerized Tomography; IOP: chronically elevated intraocular pressure >21 mm Hg resulting from damage of angle.

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
