# Peer review of "Factors Associated with Ocular and Extraocular Recovery in 143 Patients with Sarcoid Uveitis"

_jcm, 2020, doi:10.3390/jcm9123894_

Round 1

Reviewer 1 Report

None

Reviewer 2 Report

The authors did a good job in revising the manuscript according to the reviewer's suggestion

This manuscript is a resubmission of an earlier submission. The following is a list of the peer review reports and author responses from that submission.

Round 1

Reviewer 1 Report

It is very informative report describing ocular and extra-ocular recovery in sarcoid uveitis with long term observation.

  • It may be better to describe general indication of oral steroid and/or immunosuppressants for ocular manifestations associated with sarcoidosis, if possible (e.g. persistent vitritis, severe retinal vasculitis).
  • Some previous papers have suggested that efficacy of vitrectomy to subside  intraocular inflammation or reduce steroid treatment for patients with ocular sarcoidosis. Are there any cases who underwent vitrectomy in this series?
  • Are there any cases who required treatment for secondary glaucoma even after recovery? Or, does ‘recovery’ in this paper means no requirement of any kind of treatment associated with sarcoid uveitis?
  • Line 153: Table 4. What do you mean by ‘Cont’?

Reviewer 2 Report

Thank You for the opportunity to review an interesting article. There are some comments for the manuscript to be improved.

  1. Overall, the English expression needs to be improved.

  1. The median age is 58, but is described as 53 in the table 1.

  1. On page 4, line 130, '31 patients had IOP' should be replaced with 'increased IOP'.

  1. All tables are difficult to understand as a whole. Categorize it into a large topic and sub-topic, and modify it a little more clearly.

  1. In the table on page 5, what does 'j' mean at 10mg/j?

  1. In table 4, the value of systemic therapy at last report is not written.

  1. Does the 'recovery' defined here mean the recovery of 'sarcoidosis', not the recovery of 'sarcoid uveitis'? If so, the recovery of patients with other organs would be worse, but there seems to be no mention of that. In addition, about 10% of patients who used oral steroids or immunosuppressants for the treatment of extraocular sarcoidosis, presumably, were not very severe in sarcoid uveitis, I think. In the end, it is thought that extraocular involvement has a great effect on prognosis. Has this been evaluated?

Line 110: What was the reason you divided the group with age under and over 50?

Line 111: what is the definition of elevated IOP? And please explain the reason of that definition  

Table 2: What is hypertensive uveitis?

Line 143: What was the indication for systemic treatment?

Line147: What was the indication for immune suppressants treatment?

The authors need to describe the characteristics for recovery patients with another table. As we do not know what kind of value was used for the logistic regression.

Line 182: why do Caucasians recover better? Any other reasons?
